# The Catalytic Wet Oxidation of Excess Activated Sludge from a Coal Chemical Wastewater Treatment Process

Zhongquan Wang [1], Shulin Qin [1], Weicheng Zheng [1], Xiaodan Lou [1], Xu Zeng [2],[*] and Taihang Wu [2]

[1] Hangzhou Research Institute of China Coal Technology & Engineering Group, Hangzhou 311201, China; wzhquan@126.com (Z.W.); hzqsl722@163.com (S.Q.); weicheng.zheng@foxmail.com (W.Z.); lxd8982@126.com (X.L.)

[2] State Key Laboratory of Pollution Control and Resource Reuse, College of Environmental Science and Engineering, Tongji University, Shanghai 200092, China; 2232792@tongji.edu.cn

[*] Correspondence: zengxu@tongji.edu.cn; Tel.: +86-21-55088628

**Abstract:** An improved catalytic wet oxidation method for the disposal of excess activated sludge from a coal chemical wastewater treatment process by using the prepared Cu-Ce/$\gamma$-Al$_2$O$_3$ catalyst was reported. The effects of catalyst dosage, reaction temperature and time, and initial oxygen pressure on the degradation of the sludge were investigated. The maximum removal rate of volatile suspended solids, 93.2%, was achieved at 260 °C for 60 min with the catalyst 7.0 g·L$^{-1}$ and initial oxygen pressure 1.0 MPa. The removal rate of chemical oxygen demand was 78.3% under the same conditions. The production of volatile fatty acids, including mainly acetic acid, propanoic acid, and isobutyric acid, increased with the increasing temperature. These acids have the potential to be carbon sources for the biological treatment of wastewater. Scanning electron microscopy images showed that the sludge became a loose porous structure, which is beneficial for dewatering performance. The results of an energy dispersive spectroscopy analysis illustrated that the carbon element in the sludge substantially migrated from solid to liquid phases. Therefore, these results demonstrated that the proposed catalytic wet oxidation method offers a promising pathway for the disposal and utilization of excess activated sludge from the coal chemical wastewater treatment process.

**Keywords:** catalytic wet oxidation; coal chemical sludge; coal chemical wastewater

## 1. Introduction

With the fast development of the coal chemical industry, a huge amount of coal chemical wastewater has been produced. Normally, this wastewater is treated with chemical and biological treatment processes. Therefore, a massive excess of activated sludge is generated from the biological treatment plant. The sludge poses an environmental risk unless it is disposed of safely because of the contained hazardous compounds and heavy metals [1,2]. Because of the existence of hazardous materials, the biological method always showed low efficiency [3]. Incineration was chosen for the final method. However, it can produce pollutant problems, such as dioxin and toxic materials. Furthermore, incineration has high cost as a disposal method [4,5]. Nowadays, the treatment of sludge has become a difficult area for the coal chemical industry. A satisfying disposal pathway for coal chemical sludge is strongly desired.

Typically, caprolactam, an important raw material for the production of polymers, was produced massively [6]. And, the increasing trend is still ongoing because of the huge demand of humankind. In this case, a large amount of excess activated sludge was produced simultaneously [7,8]. Caprolactam is normally produced via the toluene process, the phenol process, or the benzene process. Each of them produces hazardous chemical materials, such as benzene and benzene hydrocarbon. Then, it was induced that huge amounts of hazardous materials were dissolved in wastewater [9–11]. In the biological wastewater treatment process, microorganisms will absorb a lot of these hazardous materials. Finally,

large amounts of excess activated sludge, containing huge amounts of hazardous materials, were produced. Obviously, the biological treatment process is not suitable for the treatment of wastewater. To date, the disposal of caprolactam sludge is still an urgent problem to be solved.

In recent years, advanced oxidation processes (AOPs) have attracted widespread attention due to their high efficiency in the degradation of hazardous and toxic organic contaminants [12–14]. Among them, wet oxidation (WO) exhibited a high potential for the removal of hazardous contaminants [15–18]. Because of the high oxidation activity of free radical agents, organic compounds could be oxidized to small molecular organic carboxylic acids such as acetic, formic, and oxalic acids, and even $CO_2$ and $H_2O$ [19,20]. Some commercial wet oxidation plants were established for the treatment of pharmaceuticals, textile effluents, paper mills, and olive effluents [21,22]. However, a high disposal fee is always needed for wet oxidation disposal because of its reaction conditions. Therefore, catalytic wet oxidation technology has been developed in recent years. The addition of a catalyst could enhance the reaction efficiency and decrease the operating temperature and time [23,24]. In the past few decades, transition metals have been proved to be excellent oxidizing candidates [25,26]. Specifically, Cu-based catalysts have attracted considerable attention. Li et al. reported that a Cu-loaded catalyst can accelerate the degradation of acetaldehyde [27–29]. A Cu catalyst was also reported for its efficiency in the oxidation of formic acid and phenol [30]. Furthermore, a ceria catalyst has also been proved to be efficient in wet oxidation [31,32]. Recently, a ceria-based catalyst has obtained increasing attention. However, Cu- and Ce-composite catalysts have been rarely studied.

Volatile fatty acids (VFAs), including oxalic acid, acetic acid, and propanoic acid, could be massively produced in sludge disposal [33]. For example, Gapes reported that carboxylic acids could be realized in the wet oxidation of sludge [34]. Chung reported that the propionic acid concentration increased to 13.5 mg/L at 240 °C in the wet oxidation process [35]. Wu reported on propionic and butyric acid production [36]. A lot of the available literature reported results on maximizing the yield of acetic acid. The effect of the amount of oxygen is the key influence parameter for the production of acetic acid [37]. It should be noted that the VFA solution, produced after the WO process, has the potential to be used as a kind of organic carbon source. Therefore, the production of acetic acid and VFAs through wet oxidation is well worth studying.

In this study, an improved catalytic wet oxidation method for the disposal of excess activated sludge from a coal chemical wastewater treatment process was reported by using the prepared Cu-Ce/$\gamma$-Al$_2$O$_3$ catalyst. The effects of the catalyst dosage, reaction temperature and time, and the initial oxygen pressure on the reactivity of the system were investigated. In addition, the characteristics of the produced liquid and solid were analyzed.

## 2. Results

### 2.1. Effect of Catalyst Dosage

Experiments with the catalyst addition amount from 0 to 7.0 g·L$^{-1}$ were carried out to assess the effects of the catalyst. The reaction conditions included a temperature of 260 °C and time of 60 min, with the initial oxygen pressure 1.0 MPa. The experimental results can be seen in Figure 1. As illustrated in Figure 1, the removal rates of chemical oxygen demand (COD) increased significantly in the presence of the catalyst with the amount increasing. The catalyst used in this study had a considerable catalytic effect on the wet oxidation of the sludge. In contrast, the removal rate of volatile suspended solids (VSSs) was not significant. Comparing the wet oxidation of the sludge and its oxidized intermediates, the VSS removal of the sludge occurred more easily because the removal rate of VSS was above 85% even in the absence of the catalyst, as seen in Figure 1. However, the increase in COD removal influenced the VSS removal with the addition of the prepared catalyst. When the COD removal rate was higher, the hydrolysis of the sludge was enhanced. In our previous study, the catalytic reactivity of a prepared Cu-Ce/$\gamma$-Al$_2$O$_3$ catalyst was proven to be effective for the wet oxidation of pharmaceutical sludge. It is evident that the catalytic

effect is similar for coal chemical sludge. This means that the prepared catalyst has a better capacity with the load of copper and ceria as active elements.

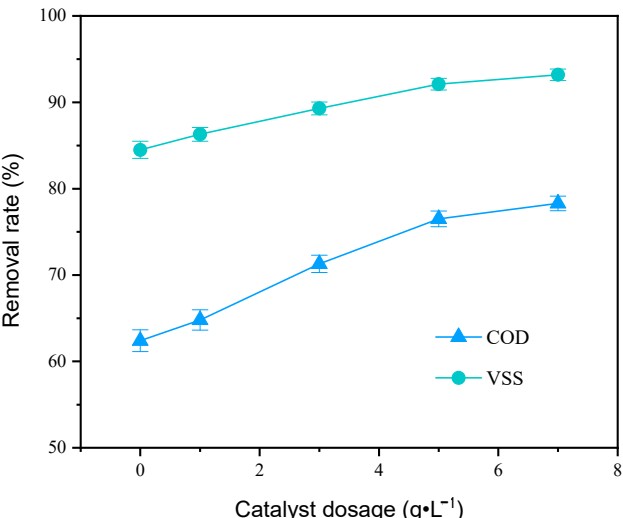

**Figure 1.** Effect of catalyst dosage (260 °C; 60 min; initial oxygen pressure 1.0 MPa).

### 2.2. Effect of Reaction Temperature

Reaction temperature always has significant influence in the wet oxidation process. On the one hand, higher temperatures provide higher reaction efficiency due to their higher energy, according to Arrhenius' law. Moreover, the solubility of oxygen gas increased when the temperature increased, which increased the oxidant for the wet oxidation of organic pollutants in the solution. In this study, the influence of temperature (180~260 °C) was investigated with a reaction time of 60 min, a catalyst of 7.0 g·L$^{-1}$, and an initial oxygen pressure of 1.0 MPa. The experimental results are shown in Figure 2.

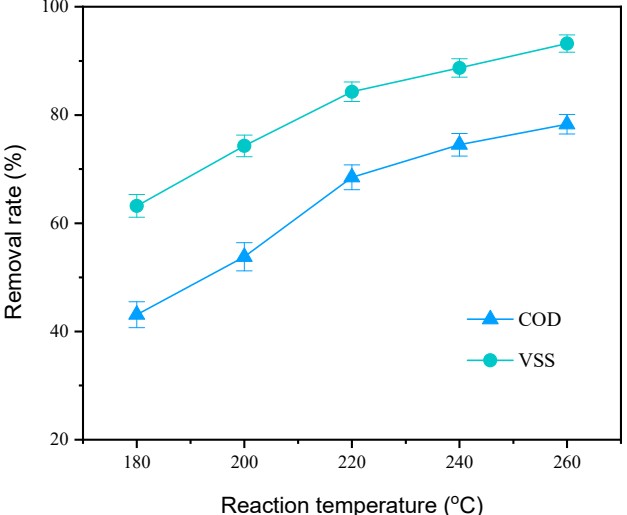

**Figure 2.** Effect of reaction temperature (60 min; initial oxygen pressure 1.0 MPa; catalyst 7.0 g·L$^{-1}$).

In Figure 2, it can be seen that the removal rates of the COD and VSSs increased significantly with the temperature increasing. Under the reaction temperature of 180 °C, the removal rates of COD and VSSs are very low. However, the removal rate of COD was almost 40%. Compared to the COD removal in the reaction, the VSS removal rate was much higher. The possible reason may be that the sludge was more easily decomposed due to hydrolysis, which happened easily under hydrothermal conditions. In the wet oxidation process, acetic acid is a common intermediate, which is not easily oxidized. Then, the

produced acetic acid could be accumulated, and the COD removal would be inhibited. Therefore, the VSS removal rate was much higher than that of the VSSs, even with surplus oxygen gas in the wet oxidation reaction system. These results were similar to Strong's results, which demonstrated that a VSS removal rate of 93% was achieved at 220 °C for 2 h under a pure oxygen pressure of 2.0 MPa. All these results imply that the reaction temperature is a key reaction parameter in the reaction. And, the reaction temperature should be high enough to acquire better removal results. However, a high temperature will increase the cost for the building and operation of the reaction system. Thus, in the industrial utilization of wet oxidation technology, the reaction temperature should be considered together with the reaction efficiency and the cost for the reaction system.

### 2.3. Effect of Reaction Time

The influence of reaction time on the sludge degradation was investigated with the experiments' reaction times varying from 20 to 60 min.

As shown in Figure 3, the COD and VSS removal rates increased simultaneously with the extension of the reaction time. A linear relationship between the increase in removal rates and the reaction time existed. The possible reason may be that the cell wall was destroyed more completely under higher temperatures during wet oxidation, which induced the solubilization of the sludge, and the decrease in the COD and VSSs when the reaction time was longer than 20 min. With the extension of the reaction time, the wet oxidation of intermediates and the hydrolysis of the sludge took place gradually. More importantly, some intermediates, which were not easily oxidized further, such as acetic acid, accumulated with the process of wet oxidation. Therefore, when the COD removal rate was near 80%, it could not increase further. Therefore, considering the oxidation efficiency and economic factors together, we chose 60 min as the favorable reaction time for the optimum reaction parameter.

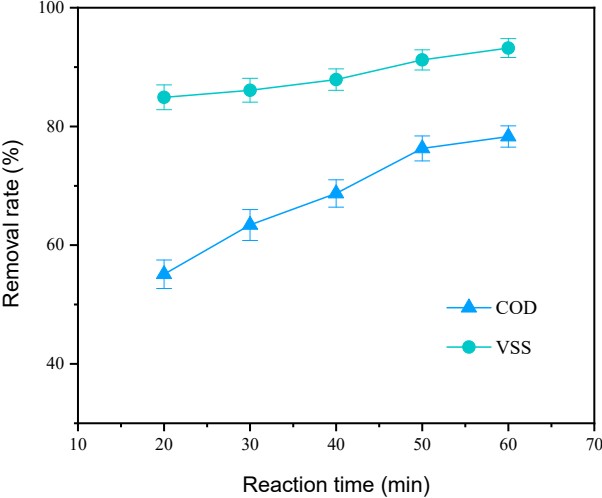

**Figure 3.** Effect of reaction time (260 °C; initial oxygen pressure 1.0 MPa; catalyst 7.0 g·L$^{-1}$).

### 2.4. Effect of Initial Oxygen Pressure

To assess the influence on the oxidant in wet oxidation, i.e., the addition of oxygen gas, experiments were carried out with the initial oxygen pressures changing from 0.2 to 1.0 MPa. The experimental results can be found in Figure 4. As seen in Figure 4, the removal of the COD was greatly influenced by the addition of oxygen gas. With the initial oxygen pressure increased, i.e., the increase in the oxidant, wet oxidation took place more easily. In contrast, the VSS removal rate was only slightly influenced by the change in the amount of oxygen. The possible reason for VSS removal could be attributed to the hydrolysis of the sludge; thus, the VSS removal rate was above 80% even with an initial oxygen pressure of 0.2 MPa. As mentioned above, the reaction temperature was the main

factor which influenced hydrolysis. Therefore, the influence of the amount of oxygen on VSS removal was small.

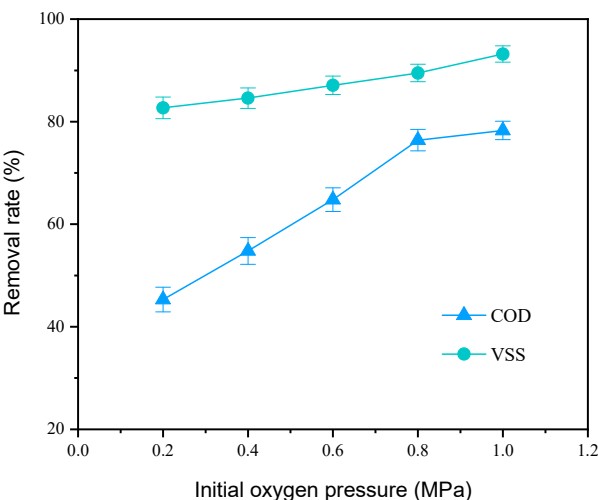

**Figure 4.** Effect of additional oxygen pressure (260 °C; 60 min; catalyst 7.0 g·L$^{-1}$).

On the other hand, the production of strong oxidation species was enhanced by the increase in additional amounts of oxygen. The dissolved amount of oxygen in the liquid increased, and the gas–liquid mass transfer speed increased. All these conditions increased the wet oxidation rate. Therefore, the COD removal rate increased significantly with the initial oxygen pressure increasing. However, the surplus addition of oxygen had a small influence on the wet oxidation of the sludge once the wet oxidation was almost stable. As shown in Figure 4, when the initial oxygen pressure increased from 0.8 to 1.0 MPa, the removal rate of the COD increased very little. A possible reason may be that the non-oxidizable intermediates were accumulated in the wet oxidation process, such as acetic acid. Therefore, the additional amount of oxygen gas was enough once it was sufficient for the removal of the COD.

### 2.5. Removal of COD and SCOD

The COD value can reflect the concentration of most organic pollutants. Therefore, the removal of organic pollutants, i.e., the degradation of the sludge, could be assessed by measuring the COD and the soluble chemical oxygen demand (SCOD). The SCOD means the organic pollutants in the liquid. As shown in Figure 5, it was found that the removal rates of the COD and the SCOD increased significantly with the increasing temperature. The maximum values were obtained at 260 °C. The difference between the COD and the SCOD is mainly due to the different reaction speeds of hydrolysis and oxidation, as the hydrolysis process goes faster than oxidation under subcritical conditions. These results also illustrated the decrease in VSSs under higher reaction temperatures, which implied that the VSSs were transferred to a liquid phase. These results illustrated that the reaction temperature should be high enough to remove the VSSs in the sludge as fast as possible.

### 2.6. Production of Acetate and VFAs

During the wet oxidation process, organic materials could be oxidized to small molecule organics, i.e., carboxylic acid and even carbon dioxide. VFAs are the dominant component of these intermediates, and they are considered to be reused as value-added materials. The effect of temperature on the concentrations of acetic acid and VFAs is shown in Figure 6.

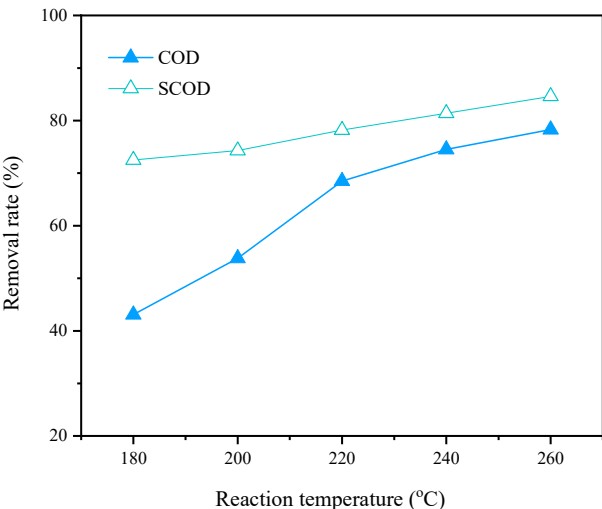

**Figure 5.** Change in COD and SCOD removal rates with different reaction temperatures.

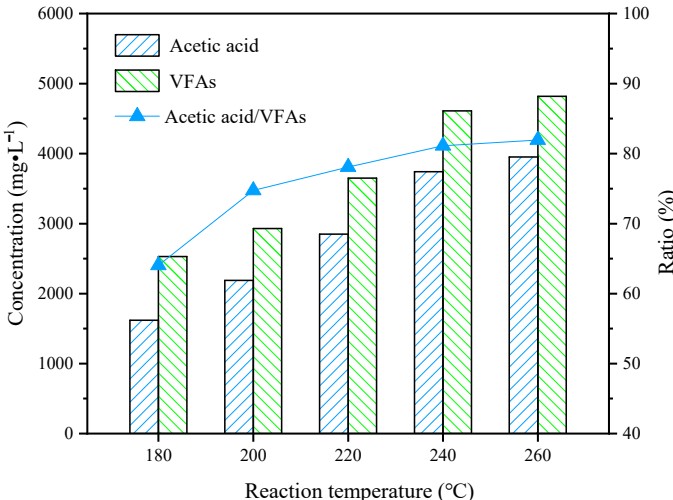

**Figure 6.** Production of acetic acid and VFAs.

As shown in the figure, the VFA concentration was found to increase dramatically while the reaction temperature rose. For acetic acid, the concentration increased from the minimum (180 °C) to the maximum (260 °C). However, it remained stable from 240 to 260 °C. It has been reported that the maximum value of the VFAs was obtained at 250 °C within 30 min, and the surplus oxidants would induce the oxidation of the VFAs, including acetic acid. Therefore, this steady state might be caused by the slight decomposition of acetic acid as the temperature was quite high and oxidizers were abundant, which reduced the formation of acetic acid from the decomposition of other organic substances. The value of acetic acid/VFAs was also calculated and it was proven that acetic acid was the main component of the VFAs. The value was quite high, almost 80%. Acetic acid dominated the post-reaction VFAs mainly because it was a stable intermediate and was relatively hard to decompose under reaction conditions. Acetate sodium was always selected as an organic carbon source in the biological wastewater treatment process. Therefore, the liquid after wet oxidation has the potential for the utilization as an organic carbon source in the wastewater treatment.

### 2.7. Characteristics of Sludge before and after WO

To investigate the changes in the microstructure and elemental composition of the sludge before and after the WO reaction, scanning electron microscopy (SEM) and energy

dispersive spectroscopy (EDS) analyses were adopted to investigate the solid after wet oxidation. Figure 7 shows the structural changes of the sludge before and after the WO reaction at the scales of 50 and 10 µm. As shown in Figure 7a,b, the sludge was in a cohesive and relatively complete block shape with sparse pores before the WO reaction. As shown in Figure 7c,d, the sludge particles became smaller and the pores increased significantly after the WO reaction. These results illustrated that the solid is conducive to the release of intercellular water and thus improves dehydration performance because of the removal of organic materials in the sludge.

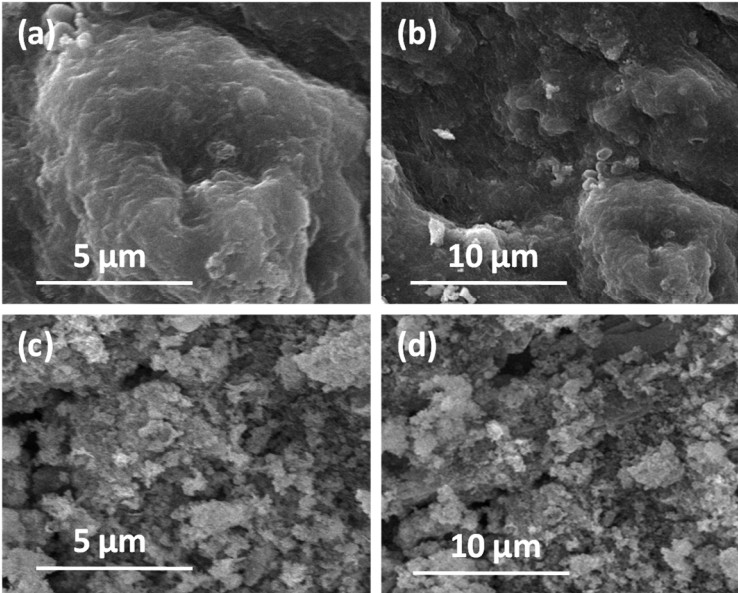

**Figure 7.** SEM images before (**a**,**b**) and after (**c**,**d**) wet oxidation (260 °C, 1.0 MPa O$_2$, pH = 8.09; left to right: 100 nm, 500 nm, and 1 µm).

Figure 8 shows the energy spectrum analysis results. As shown in Figure 8, the changes in the content of the carbon and oxygen elements in the sludge after the WO reaction were significantly decreased compared with the results before the WO reaction. The carbon content in the sludge decreased by almost 30% between the results from before and after the reaction, indicating the removal of organic substances. The VSS removal rate was above 90% at 260 °C for 60 min, which means that the amount of solids decreased significantly after the wet oxidation. Based on the calculations, the amount of metals increased. The possible reason may be that the metal ions were oxidized and precipitation happened. Therefore, wet oxidation can not only oxidize the organic pollutants, but it can also remove some heavy metals by changing the pH of the liquid.

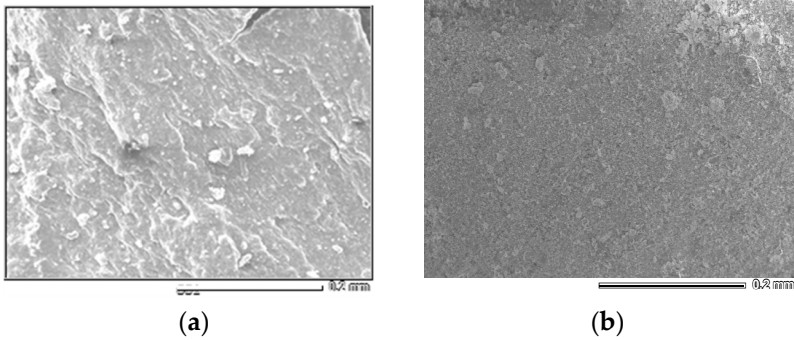

(**a**)　　　　　　　　　　　　(**b**)

**Figure 8.** EDS images before (**a**) and after (**b**) the sludge WO reaction.

## 3. Discussion

It is very hard to illustrate the distinct mechanism because the sludge is very complex, and the intermediates were also very complex. The sludge consisted of microorganisms and organic pollutants, but the detailed mechanism of sludge wet oxidation is still unknown [38,39]. However, the wet oxidation reaction is always assumed as a free radical oxidation reaction [40]. Normally, wet oxidation was assumed to be a free radical reaction mechanism. Under hydrothermal conditions, large amounts of free radicals were produced based on the dissolution of oxygen, which possessed strong oxidant activity. Then, the oxidation reaction took place easily, and the pollutants were oxidized and removed.

For the wet oxidation of the sludge, based on the experimental results and discussion above, it is supposed that hydrolysis was the first step. In the hydrolysis process, proteins and lipids decomposed into a liquid, along with the dissolution of inorganic materials. Then, a high VSS removal rate was achieved. Because the reaction temperature was very high, the hydrolysis process took place very quickly. Once the solid materials were transferred to a liquid phase, the compounds were oxidized by the strong oxidant free radicals. Then, a lot of VFAs were produced, including acetic acid, isobutyric acid, and propanoic acid, etc. [41,42]. The $BOD_5/COD$ ratio of the liquid was very high, even above 0.70. In some industries, acetate sodium was always bought as a carbon source for industrial refractory wastewater treatment. Some experimental results proved that the oxidation liquid is useful for nitrogen removal in wastewater treatment. Therefore, the liquid after wet oxidation has the potential to be utilized as an organic carbon source in the biological wastewater treatment process. The advantages of WO also include exothermic reactions, which can be used to amend the energy consumption to maintain the occurrence of the reaction. After the WO reaction, the sludge particles became smaller and the pores significantly increased, which was conducive to the release of intercellular water and thus improved the dehydration performance. The solid could be used for the production of ceramic aggregate or permeable brick. Therefore, WO can be regarded as an ideal method for the volume reduction and resource utilization of pharmaceutical sludge.

In the present study, wet oxidation was conducted in a batch reactor. After the catalyst was used several times, the enhancement of the catalyst for the COD and VSS removal was still above 95%. And, the catalyst became more stable when the catalyst was used for wet oxidation for a long time. Therefore, for the industrial utilization of this method, there is still a need to study it in a continuous reaction system. In addition, the stability of the catalyst still needs to be verified. Under the hydrothermal reaction conditions, the leaching of the active elements, including copper and ceria, should be discussed. Furthermore, the catalyst should be developed further. Once the cost for the wet oxidation process is accepted for commercial utilization, the catalytic wet oxidation method could be broadly industrially utilized.

## 4. Materials and Methods

### 4.1. Materials

The excess activated sludge used for the experiments was obtained from a coal chemical industry in China, which produced caprolactam. The initial parameters of the sludge were the following: COD 16,500~17,500 mg·$L^{-1}$, VSS/SS 81.3%, pH, 8.47. The sludge was stored in a refrigerator at 4 °C to avoid possible biological activities. All the reagents used in this study were analytical grade without any treatment.

### 4.2. WO Reaction System

WO runs were performed in a stainless-steel autoclave reactor with an inner volume of 200 mL. The functions of heating, stirring, and temperature monitoring were integrated and equipped in the reactor. The information on the reaction system can be found elsewhere [43]. A diagram of the reaction system is shown in Figure 9. For a typical experiment run, 100 mL of the sludge solution was added to the reactor initially. Then, oxygen gas at 0.5 MPa was purged and vented three times to remove the air inside. On the fourth purge, oxygen was

stored inside the reactor as the oxidant and the pressure was increased to 0.2~1.0 MPa. The reaction temperature was set between 180 and 260 °C and monitored using a thermocouple to maintain a deviation of 2 °C. Once the reactor achieved the set temperature, the reaction time would start to count and was set to 20~60 min. The stirrer speed was kept constant at 300 rpm throughout the process. The experiments for each reaction condition were carried out three times. As the experiments were conducted in batch mode, the inner pressure was self-pressurized. When the reaction finished, the reaction was moved and cooled to room temperature. The liquid and solid were collected and sampled.

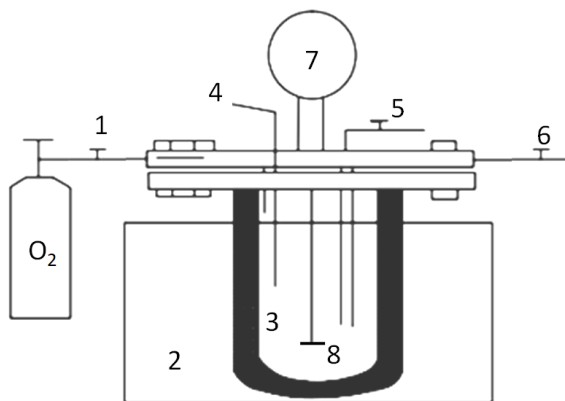

**Figure 9.** Diagram of wet oxidation reactor; 1—oxygen supply, 2—heating jacket, 3—oxidation reactor, 4—thermocouple, 5—discharge of gas, 6—bleeder valve, 7—pressure meter, and 8—stirrer.

### 4.3. Preparation of Catalyst

Detailed information about catalyst preparation and characteristics can be found elsewhere [44]. The catalyst was prepared using an impregnation process. The solution of $Cu(NO_3)_2$ and $Ce(NO_3)_3$ with a molar ratio of 1:1 and the concentration 1.0 mol·$L^{-1}$ was prepared. $\gamma$-$Al_2O_3$, as a carrier, was put into the solution. After 24 h of impregnation, the carrier was collected, and then it was baked at 550 °C for 4 h. Finally, the Cu-Ce/$\gamma$-$Al_2O_3$ catalyst was prepared.

### 4.4. Analysis Method

Various parameters such as the COD, SS, VSSs, and pH were determined according to standard methods [45]. After acidification (pH < 2) and filtration with a 0.22 μm membrane, the VFAs were identified and quantitatively analyzed by using gas chromatography (GC, Persee G5, Beijing, China) with bonded polyethylene glycol capillary columns (DB-FFAP, Agilent, Santa Clara, CA, USA) and a flame ionization detector (FID). Helium gas (Huaxiong Gas Co., Ltd., Jiaxing, China) was selected as the carrier gas. The powdered sample of dried sludge and WO residue were characterized using SEM (Gemini 300, ZEISS, Oberkochen, Germany) and EDS (INCA X-Act, Oxford, United Kingdom) to observe the morphology and element distribution.

### 5. Conclusions

In the present study, an improved catalytic wet oxidation method for the disposal of excess activated sludge from a coal chemical wastewater treatment process was reported by using the prepared Cu-Ce/$\gamma$-$Al_2O_3$ catalyst. The effects of catalyst dosage, reaction temperature and time, and initial oxygen pressure on the degradation of the sludge were investigated. The highest removal rate of volatile suspended solids, 93.2%, was achieved at 260 °C for 60 min with the catalyst 7.0 g·$L^{-1}$ and an initial oxygen pressure of 1.0 MPa. The removal rate of the chemical oxygen demand was 78.3% under the same conditions. The concentration of volatile fatty acids, including mainly acetic acid, propanoic acid, and isobutyric acid, increased with the temperature increase. The trend of VFAs and acetic acid demonstrated that there was a peak in the promotion of organic acid content by

increasing the temperature. These acids have the potential to be carbon sources for the biological treatment of wastewater. Our previous study proved that nitrogen removal could be enhanced with the addition of a wet oxidation liquid in the inlet wastewater in an A/O wastewater treatment system. Scanning electron microscopy images showed that the sludge became a loose porous structure, which is beneficial for the dewatering performance. The results of an energy dispersive spectroscopy analysis illustrated that the carbon element in the sludge substantially migrated from solid to liquid phases. The utilization of the solid could be realized through the production of ceramic aggregate. Future work should be completed for industrial utilization. These results demonstrated that the catalytic wet oxidation process provides a suitable alternative pathway for the disposal and utilization of excess activated sludge from the coal chemical wastewater treatment process.

**Author Contributions:** Conceptualization, X.Z. and S.Q.; methodology, W.Z.; software, T.W.; validation, Z.W.; formal analysis, T.W.; investigation, X.Z.; writing—original draft preparation, S.Q.; writing—review and editing, X.Z. and X.L.; supervision, X.Z.; project administration, S.Q.; funding acquisition, S.Q. All authors have read and agreed to the published version of the manuscript.

**Funding:** This work was financially supported by the Science and Technology Innovation Project of the China Coal Technology & Engineering Group (No. 2022-3-KJHZ001).

**Data Availability Statement:** The data presented in this study are available on request from the corresponding author.

**Conflicts of Interest:** The authors declare no conflict of interest.

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
