# Peer review of "The Catalytic Wet Oxidation of Excess Activated Sludge from a Coal Chemical Wastewater Treatment Process"

_catalysts, doi:10.3390/catal13101352_

Round 1
Reviewer 1 Report
The manuscript entitled “Catalytic wet oxidation of excess activated sludge from a coal chemical wastewater treatment process” was reviewed. In general, the manuscript quite interesting. However, most parts appear to be not prepared well.
Introduction: a) Lines 26/36 (1st paragraph): Add references; b) Lines 37/39: Add references.; c) Lines 49/51: I recommend add a recent reference (d)
https://doi.org/10.1016/j.psep.2021.09.029).; d) How is this system different to other reports to merit publication? Please, report.
Results: The experimental data should be represented as average ± standard deviation. The authors should also report the number of replicates for all assays.
Materials and methods: a) The authors should add relevant information on the reactor and experimental conditions (ratio height/diameter, volume, stirring or static conditions; stirring frequency). I recommend add an illustrative scheme of the experimental apparatus used in this study (reactor). It is very important to the readers.
The manuscript entitled “Catalytic wet oxidation of excess activated sludge from a coal chemical wastewater treatment process” was reviewed. In general, the manuscript quite interesting. However, most parts appear to be not prepared well.
Introduction: a) Lines 26/36 (1st paragraph): Add references; b) Lines 37/39: Add references.; c) Lines 49/51: I recommend add a recent reference (d)
https://doi.org/10.1016/j.psep.2021.09.029).; d) How is this system different to other reports to merit publication? Please, report.
Results: The experimental data should be represented as average ± standard deviation. The authors should also report the number of replicates for all assays.
Materials and methods: a) The authors should add relevant information on the reactor and experimental conditions (ratio height/diameter, volume, stirring or static conditions; stirring frequency). I recommend add an illustrative scheme of the experimental apparatus used in this study (reactor). It is very important to the readers.
Reviewer 2 Report
The manuscript by Wu and coworkers, titled “Catalytic wet oxidation of excess activated sludge from a coal chemical wastewater treatment process”, reports on the use of a Cu-Ce/γ-Al2O3 catalyst for degradation of sludge from a coal chemical wastewater treatment process. Several parameters were investigated aiming at optimizing the degradation process.
The work could be of interest for the readers of catalysts, after the following major revision.
1) Recyclability studies with the same catalytic systems are highly recommended;
2) English needs to be polished;
3) Please, specify acronymes the first time they are mentioned.
English needs to be polished
Round 2
Reviewer 2 Report
The authors adequately responded to all referees' concerns. Therefore, I suggest publication in the present form